# PD-L1 Expression Is Significantly Associated with Tumor Mutation Burden and Microsatellite Instability Score

**DOI:** 10.3390/cancers13184659

**Published:** 2021-09-16

**Authors:** Yoon Ah Cho, Hyunwoo Lee, Deok Geun Kim, Hyunjin Kim, Sang Yun Ha, Yoon-La Choi, Kee-Taek Jang, Kyoung-Mee Kim

**Affiliations:** 1Samsung Medical Center, Department of Pathology and Translational Genomics, Sungkyunkwan University School of Medicine, Seoul 06351, Korea; purpleice21@hallym.or.kr (Y.A.C.); hwpatho.lee@samsung.com (H.L.); h2407.kim@samsung.com (H.K.); sangyun.ha@skku.edu (S.Y.H.); ylachoi@skku.edu (Y.-L.C.); ktjang12@gmail.com (K.-T.J.); 2Department of Pathology, Hallym University Sacred Heart Hospital, Hallym University College of Medicine, Anyang 14068, Korea; 3Department of Clinical Genomic Center, Samsung Medical Center, Seoul 06351, Korea; deokgeun.kim@samsung.com; 4Department of Digital Health, Samsung Advanced Institute of Health Science and Technology, Sungkyunkwan University, Seoul 06351, Korea

**Keywords:** comprehensive cancer panel assay, tumor mutation burden, PD-L1 score, microsatellite instability

## Abstract

**Simple Summary:**

Biomarkers for predicting the response to immune checkpoint blockade (ICB) includes programmed death-ligand 1 (PD-L1) immunohistochemistry (IHC), microsatellite instability (MSI), and tumor mutation burden (TMB). This study investigated the relationship of these biomarkers using comprehensive cancer panel assay (CCPA) with >500 genes in 588 advanced cancer patients. The work demonstrates that PD-L1 expression is significantly associated with TMB and MSI score, according to primary tumor origin.

**Abstract:**

Programmed death-ligand 1 (PD-L1) immunohistochemistry (IHC), microsatellite instability (MSI), and tumor mutation burden (TMB) have been proposed as a predictive biomarker to predict response to immune checkpoint blockade (ICB). We aimed to find the relationship of PD-L1 IHC to TMB and MSI using a comprehensive cancer panel assay (CCPA) with >500 genes in advanced cancer patients. CCPA results from 588 archived tissue samples were analyzed for TMB and MSI. In seven samples, whole exome sequencing confirmed TMB with Pearson’s correlation coefficient of 0.972 and all MSI-high cases were validated by pentaplex PCR. Association of TMB and MSI with their corresponding PD-L1 IHC was analyzed. The median TMB value of 588 cases was 8.25 mutations (mut)/Mb (range 0–426.8) with different distributions among the tumor types, with high proportions of high-TMB (>10mut/Mb) in tumors from melanoma, colorectal, gastric, and biliary tract. The TMB values significantly correlated with PD-L1 expression, and this correlation was prominent in gastric and biliary tract cancers. Moreover, the MSI score, the proportion of unstable MSI sites to total assessed MSI sites, showed a significant correlation with the TMB values and PD-L1 scores. This study demonstrates that PD-L1 expression is significantly associated with TMB and MSI score and this correlation depends on the location of the primary tumor.

## 1. Introduction

Identifying targetable alterations and biomarkers for predicting immunotherapeutic response with limited specimens is crucial [1]. Recent advances in precision oncology have promoted the widespread use of, and interest in, developing cost-effective cancer panel assays to accurately measure microsatellite instability (MSI) and tumor mutation burden (TMB) for predicting responses to immunotherapy.

Immune checkpoint inhibitors (ICI) promote host attack on cancer cells by engaging in the programmed death-ligand 1 (PD-L1)-programmed death 1 (PD-1) pathway and have been shown to have an impressive effect against several types of solid organ cancers, including non-small-cell lung cancer [2,3,4,5]. ICI has been approved by the U.S. Food and Drug Administration for various tumor types under various conditions [6]. To enhance ICI therapeutic efficacy, the evaluation of biomarkers for proper patient selection is necessary [7,8]. The initial biomarker used to predict the therapeutic response to ICI was the expression of PD-L1. The expression of PD-L1 on tumor cells or tumor-infiltrating immune cells by immunohistochemistry (IHC) has become the most widely used biomarker for selecting patients for ICI therapy [9]. Recently, TMB, defined as the total number of mutations per megabase (mut/Mb) in the coding area of a tumor genome, has emerged as a predictive biomarker for response to ICI [7,10,11,12]. TMB was initially measured through whole-exome sequencing (WES) by calculating the number of somatic mutations per Mb [13,14,15,16,17,18,19], which is costly and time-consuming [15,16,17]. Highly mutated tumors are more likely to produce abundant tumor-specific mutant epitopes functioning as neoantigens and are targeted by the immune system [15,20].

MSI, the condition of genetic hypermutability caused by the inactivation of mismatch repair genes, is a known biomarker for immunotherapy [21,22]. Detection of MSI is important for cancer prognosis, therapy strategy, and extracting information regarding familial cancer risk [23,24,25,26]. Polymerase chain reaction (PCR) followed by capillary electrophoresis fragment analysis using a relatively small standardized panel of microsatellites is the gold standard for MSI testing [27,28,29]; however, MSI detection using NGS has also recently emerged [29,30,31]. Furthermore, a direct comparison of MSI PCR and pan-cancer panels is needed for MSI detection.

In this study, to determine the relationship of biomarkers for patient stratification in the immunotherapy of patients with advanced solid tumors, we compared PD-L1 IHC results with TMB values and the percentage of unstable microsatellite loci using a commercially available comprehensive cancer panel assay (CCPA) with >500 genes.

## 2. Materials and Methods

### 2.1. Patients and Tumor Samples

Five hundred and eighty-eight patients who received systemic chemotherapy at Samsung Medical Cancer between August 2017 and October 2020 were included in this study. The tumor samples were formalin-fixed paraffin-embedded (FFPE) tissues, and tumor DNA was obtained by microdissection of the tumor regions for most samples, except for small samples for CCPA. All procedures were performed in a CAP-accredited laboratory (Samsung Medical Center, Seoul, Korea). For WES, DNA from fresh tumor tissues and their blood were used. The average age of patients was 58.26 years (range, 18–87 years); 358 (60.9%) patients were men. Almost half of the patients’ tumors were resected (50.9%). Most specimens were acquired from our institute (90.8%), with an average paraffin age of 250.72 days (range, 2–3536). The tumors included carcinomas from the gastrointestinal, hepatobiliary, genitourinary tracts, adrenal cortical carcinoma, malignant melanoma (MM), neuroendocrine tumor, and mesenchymal tumors. The tissues were mainly of primary origin (81.3%). The clinicopathological characteristics of the patients and the methods of specimen acquisition are shown in Table 1.

Written informed consent for genetic analysis and the use of clinical data records were obtained from all patients. The study was approved by the Institutional Review Board of the Samsung Medical Center (IRB number: 2020-06-045-002).

### 2.2. DNA Extraction

Genomic DNA was extracted from FFPE tissue sections (generally 6–10 mm in size) and purified using a Qiagen AllPrep DNA/RNA FFPE Kit (Qiagen, Venlo, The Netherlands). Qubit dsDNA HS Assay Kit (Thermo Fisher Scientific, Waltham, MA, USA) was used for quantitating DNA concentration, and 120 ng of input DNA was used for library preparation after modification of the manufacturer’s instructions. We used Genomic DNA ScreenTape (Agilent Technologies, Santa Clara, CA, USA) on an Agilent 2200 TapeStation system (Agilent Technologies, Santa Clara, CA, USA) to assess integrity, fragment size, and quality of DNA.

### 2.3. Library Preparation, Sequencing, and Data Analysis

The library was prepared according to the manufacturer’s protocol using a hybrid capture-based TruSight Oncology 500 DNA/RNA NextSeq Kit (Illumina, San Diego, CA, USA). During library preparation, enrichment chemistry was optimized to capture nucleic acid targets from FFPE tissues. In the TSO 500 analysis, unique molecular identifiers were used to determine the unique coverage at each position and reduce the background noise caused by sequencing and deamination artifacts in the FFPE samples. This simultaneously suppresses errors and enables the detection of variants at low variant allele frequencies (VAFs), resulting in high specificity during DNA library preparation.

Sequence data were analyzed for clinically relevant classes of genomic alterations, including single nucleotide variants (SNVs), small insertions and deletions (indels), copy number variations (CNVs), and rearrangements/fusions. Results of SNVs and small indels with a VAF of less than 2% were excluded. Average CNVs of more than 4 copies were considered as “gain,” and less than one was considered as “loss.” Only gain (amplification) was analyzed in the TSO 500 CNV analysis, and RNA translocation supporting reads more than 4 to 12 were considered a translocation, depending on the quality of the sample. Data outputs exported from the TSO 500 pipeline (Illumina, San Diego, CA, USA) [32] were annotated using the Ensembl Variant Effect Predictor (VEP) Annotation Engine [32], with information from databases such as dbSNP, gnomAD genome, 1000 genomes, ClinVar, COSMIC, and RefSeq (NCBI Reference Sequence Database version VEP build 91, Bethesda, MD, USA) The processed genomic changes were categorized according to a 4-tier system proposed by the American Society of Clinical Oncology and the College of American Pathologists [33]. TMB was defined as the number of somatic mutations per Mb, and ≥10 was defined as high TMB and <10 was classified as low TMB [13,14,15,16,17,18,19], calculated by (1) excluding any variant with an observed allele count ≥10 in any of the GnomAD exome, genome, and 1000 genomes databases, and including (2) variants in the coding region (RefSeq Cds; NCBI Reference Sequence Database version VEP build 91, Bethesda, MD, USA), (3) variant frequency ≥5%, (4) coverage ≥50×, (5) SNVs and indels, and (6) non-synonymous and synonymous variants. The effective panel size for TMB is the entire coding region with a coverage of >50×. Calculation of the TMB is summarized in Appendix A. The MSI status is calculated from microsatellite sites for evidence of instability relative to a set of baseline normal samples that are based on information entropy metrics. The final NGS study results were reviewed by three pathologists (Y.A.C., H.L. and K.M.K.) and a bioinformatics expert (D.G.K.).

### 2.4. Immunohistochemistry

IHC analysis of whole representative FFPE tissue blocks was performed for PD-L1 expression using the PD-L1 22C3 pharmDx assay (Agilent Technologies). Membranous staining for PD-L1 was considered PD-L1-positive, and PD-L1 expression status was calculated as the percentage of PD-L1-positive cells of the total tumor cells. PD-L1 TPS and CPS were categorized by either a three- or two-tier system, according to previous reports [34,35].

### 2.5. MSI Analysis

In all cases having high MSI (microsatellite instability) with ≥20 unstable MS loci by TSO500 results were defined as MSI-high. The proportion of unstable MSI sites to total assessed MSI sites is reported as a sample-level microsatellite score, in which at least 40 sites were required to determine a MSI score. All MSI-high cases were confirmed with multiplex PCR with five quasi-monomorphic mononucleotide repeat markers as previously described [36]. Briefly, we used a QIAamp DNA Mini kit (Qiagen, Venlo, The Netherlands) for genomic DNA extraction from FFPE tumor tissues or unstained slides. Each PCR primer was labeled at the 5′ end with one fluorescent FAM, HEX, or NED marker. Amplified PCR products were then analyzed on an AB 3500xL Dx Genetic Analyzer (Applied Biosystems, Foster City, CA, USA). A Genemapper 4.1 (Applied Biosystems, Foster City, CA, USA) device was used to estimate allele sizes. Tumors with allele size variations in two or more of the microsatellite markers were classified as MSI-high (MSI-H) according to the National Cancer Institute guidelines to determine MSI in colorectal cancer [27].

### 2.6. Statistical Analysis

The correlation between alteration and TSO 500 results was calculated using Spearman’s coefficient using SPSS ver. 25.0 software (IBM Corp., Armonk, NY, USA). To analyze correlation between TMB, MSI and PD-L1 expression, chi-squared test and Fisher’s exact test were used. The t-test or Wilcoxon rank-sum test was used to compare continuous variables. Two-sided *p*-values < 0.05 were considered statistically significant for all analyses. Visualization of genetic alterations was analyzed using the GenVisR R-package (Bioconductor, version 3.6.3) [37].

## 3. Results

### 3.1. Overall Sequencing Quality

The overall sequencing quality and recommended guideline quality thresholds are presented in Appendix A. We compared the average depths of sequencing coverage obtained with the bead pooling method (Appendix A) and manual pooling method (Appendix A) to assess the sequencing depths and found that the manual pooling method (sequencing coverage depth, 1724.4) was superior to the bead pooling method (sequencing coverage depth, 1655.7). The average depth of sequencing coverage was ×150–300, when 40 ng input of DNA, which is the minimum amount of the manufacturer’s guidelines, was used. To ensure stable and reliable depth coverage, we altered the amount of DNA input. When we increased the amount of DNA input from 80 ng (sequencing coverage depth, 1724.4) to 120 ng, the average depth coverage (sequencing coverage depth, 2041.5) also increased to 317.1× (Appendix A). Finally, we simultaneously tested the sequence coverage, duplication rate, and insert sizes to compare the inter-institutional variation of the sample source (FFPE from authors and referred from other hospitals). However, there was no significant difference in coverage (*p* = 0.891), level of duplication (*p* = 0.24), or insert size (*p* = 0.879) between the institutions.

### 3.2. Measurement of TMB

The median TMB value of 588 cases was 8.25 mutations/Mb (range 0–426.8). TMB distribution varied according to tumor types; cancer of the small intestine (SI) showed the highest average TMB (36.31 ± 112.42 mut/Mb), followed by cancer of the female genital tract (15.36 ± 11.37 mut/Mb), lung cancer (11.51 ± 17.42 mut/Mb), cancer of the biliary tract (9.42 ± 20.21 mut/Mb), colorectal cancers (CRC) (9.37 ± 19.07 mut/Mb), cancer of the genitourinary tract (GU) (8.95 ± 6.66 mut/Mb), gastric cancers (GC) (6.81 ± 7.68 mut/Mb), cancer of the liver, except cholangiocarcinoma (6.25 ± 1.95 mut/Mb), MM (5.40 ± 4.33 mut/Mb), pancreas cancers (4.80 ± 2.34 mut/Mb), gastrointestinal stromal tumor (GIST) (4.01 ± 1.42 mut/Mb), and mesenchymal tumor (3.69 ± 2.13 mut/Mb) (Figure 1A). When high-TMB was defined as more than 10 mut/MB, the proportions of high-TMB were 34.6%, 21.8%, 18%, 7.7%, 5.1%, and 3.8% in the CRC, GC, biliary tract, GU, female genital tract and respectively (Figure 1B). For comparison, the proportion of TMB was analyzed using previously reported TCGA data (Figure 1C) [38,39]. Previous results have shown the highest TMB proportions in melanoma, CRC, GC, and CC. Tumor volume in the 588 cases ranged from 10% to 95%, with an average of 53.08%. However, tumor volume was not significantly correlated with TMB (Spearman’s correlation coefficient = 0.067 and *p* = 0.103; Figure 1D).

### 3.3. Validation and Correlation of TMB between TSO 500 and WES

We collected fresh tumor samples from seven patients for the validation of the TMB values. Six patients had advanced CRC, and one patient had MM. The mean age was 54.57 years (range, 20–70 years). Four patients (57.1%) were male, and the mean age of the paraffin block was 236.14 days (range, 26–1175).

The total size of the coding region was 1.28 Mbps in TSO 500 and 35.9 Mbps in WES. The average reading coverage depth in the TSO 500 was 713.29 (median, 522; range, 445–1525), which was higher than the 150× recommended by the user’s guideline, and the percentage of coding regions covering 50× was 99.6% or more, all of which were adequate enough to interpret the data. The TMB values of the TSO 500 and WES are summarized in Appendix A. The overall correlation of the TMB values between the TSO 500 and WES results are shown in Appendix A. The Pearson’s correlation coefficient was 0.972.

### 3.4. Correlation of PD-L1 with TMB and MSI

Among the 588 cases, 413 cases were available to compare the PD-L1 score with TMB values and MSI. The TMB value was significantly correlated with the PD-L1 CPS and TPS (Pearson correlation coefficient = 0.112 and 0.100, *p* = 0.022 and *p* = 0.043, respectively; Figure 2A,B). After subdividing the tumor group by the primary organs, the TMB value of GC showed the most significant correlation with PD-L1 CPS (Pearson correlation coefficient = 0.513, *p* < 0.001; Figure 2C) and PD-L1 TPS (Pearson correlation coefficient = 0.625, *p* < 0.001; Figure 2C,D). Biliary tract carcinoma (extrahepatic and cholangiocarcinoma) also showed high correlations (Pearson correlation coefficient = 0.362 and 0.427, *p* = 0.002 and *p* < 0.001, respectively; Figure 2E,F). However, the TMB values in other organs failed to show statistical significance.

PD-L1 TPS and CPS were categorized as either three- or two-tier systems, according to previous reports [35,36]. TMB was categorized as low or high TMB with a cut-off value of 10 mut/Mb. When PD-L1 CPS was categorized as three-tier with 1 and 50 as cut-off values, high TMB was significantly associated with high PD-L1 CPS in all cases (*p* = 0.005), gastric cancers (*p* = 0.001), and biliary tract cancers (*p* = 0.032; Table 2). When applying the two-tier criteria of PD-L1 CPS cut-off, PD-L1 positivity was significantly associated with high TMB in all cases (*p* = 0.017; Table 2) and showed a tendency toward high TMB in gastric cancers (*p* = 0.060; Table 2). However, unlike PD-L1 CPS, only gastric cancer showed a significant correlation between PD-L1 TPS and TMB (*p* < 0.001; Table 3).

The MSI score provided by the TSO 500 showed a significant correlation with TMB values (Pearson correlation coefficient = 0.377, *p* < 0.001, Figure 3A). In addition, MSI score significantly correlated with the PD-L1 CPS score (Pearson correlation coefficient = 0.160, *p* = 0.001; Figure 3B) and the PD-L1 TPS (Pearson correlation coefficient = 0.154, *p* = 0.002, Figure 3C). This correlation was especially prominent in biliary tract carcinoma and GC. In biliary tract carcinomas, MSI scores were significantly correlated with the TMB values (Pearson correlation coefficient = 0.694, *p* < 0.001) and PD-L1 CPS (Pearson correlation coefficient = 0.285, *p* = 0.014, Figure 3D). However, MSI scores were not correlated with the PD-L1 TPS (Pearson correlation coefficient = 0.185, *p* = 0.118; Figure 3E). In GC, the MSI scores were significantly correlated with the TMB values (Pearson correlation coefficient = 0.918, *p* < 0.001), PD-L1 CPS (Pearson correlation coefficient = 0.540, *p* < 0.001, Figure 3F), and the PD-L1 TPS (Pearson correlation coefficient = 0.711, *p* < 0.001; Figure 3G). Even with the low incidence of MSI-H cases (8/414) in our cohort, the incidence of MSI-H was higher at 15.7% (8/51) in the high TMB group compared to 0% (0/363) in the low TMB cohort (*p* < 0.001; Table 4). In addition, the prevalence of MSI-H was associated with PD-L1 three-tiered TPS (*p* < 0.001; Table 4) and a tendency toward two-tiered PD-L1 CPS (*p* = 0.091; Table 4).

## 4. Discussion

With the advent of the molecular era, identifying suitable biomarkers has become the most important consideration of the treatment plan. The evaluation PD-L1 expression, MSI and TMB is important in predicting ICI response and can provide valuable information to many clinicians. Highly mutated tumors are more likely to produce abundant tumor-specific mutant epitopes, which may function as neoantigens [15,20] and upregulates PD-L1 expression [40,41]; however, a large study reported the opposite [34]. In this study, we attempted to evaluate the association of PD-L1 expression with TMB and MSI.

The gold standard for TMB calculation is WES, which is costly and time-consuming [15,16,17]; however, several studies have reported that targeted NGS also represent the results of WES [32,42]. Furthermore, measuring TMB is easier than evaluating neoantigen production, considering that TMB value is correlated with the presence of tumor neoantigens [20,43,44]. Our study also conducted WES to compare the results of TSO 500 with a panel size of 1.8 Mbp and found that TSO 500 is relatively free from bias induced by the panel size [45]. For the proper assessment of TMB, evaluation of DNA input, DNA quality, and tumor cell percentage are also important [1]. In the present study, we used a total of 120 ng of genomic DNA, which is higher than minimal recommendation (40 ng) [1] and we identified that tumor volume was not significantly correlated with TMB. Therefore, when sufficient genomic DNA is guaranteed, TSO 500 can be used to evaluate the TMB with a relatively low tumor volume.

In this study, we selected seven tumor samples for WES to validate the TMB results of the TSO 500. Although both results showed a higher correlation, however, we could identify that germline variants being filtered in the WES are not sufficiently filtered out in the tumor-only sequencing with TSO 500. This, in turn, may have occurred due to the limitation of the targeted NGS panel that analyzes tumor DNA without paired normal tissue where germline variants cannot be effectively filtered out compared to that in WES [46]. It is crucial to find a way to filter germline variants more effectively by tumor tissue analyses, such as those performed after the robust update of germline databases.

With this study, we found that the median value of TMB was different among the various tumors; the site with the highest TMB value was from the tumors from small intestine and female genital tract due to several outlier cases with very high TMB values exceeding 400. When those cases were excluded, the lung and biliary tract cancers showed high median TMB values, followed by CRC, genitourinary tract, GC, liver cancer, MM, pancreatic cancer, GIST, and mesenchymal tumor. Melanoma is a tumor known to show a high-TMB [34,39]; however, in the present study, the TMB of melanoma was lower than that of gastrointestinal tumors. This might be due to our patient cohort since the TMB of melanoma differs among MM subtypes [47]. Mucosal, uveal, and acral lentiginous-type melanoma, which corresponded to 13 out of 14 MM cases, showed a lower TMB than other subtypes [47], so the TMB of melanoma in the present study seems to be lower than that of previous studies [47,48]. In addition, the median TMB value of biliary tract cancer was also higher than that of other tumor types, even though we included intrahepatic cholangiocarcinoma (CCC). This might be caused by different risk factors due to geographic differences compared with Western countries, where incidence is low and inflammatory bowel disease, primary cholangitis, and hepatitis C are major risk factors [49]. However, the incidence of intrahepatic CCC is high in Asia, and the major risk factors are liver fluke infection and hepatitis B infection, which are more associated with chronic inflammatory conditions [50]. A study indicated a genetic discrepancy between Western and Asian CCC, in which Asian CCC showed a higher frequency of high-TMB than Western CCC [51].

In this investigation with cancers from various primary sites, the TMB values correlated significantly with the PD-L1 CPS score, and this significance remained when the TMB was categorized as low or high. Furthermore, the strength of correlation between PD-L1 expression and TMB may differ among tumor types, with a high correlation found in GCs and endometrial cancers and a weak correlation with renal cell carcinoma, pancreatic cancer, and MM [34,52]. In our study, GC showed the most significant correlation with PD-L1 CPS and TPS scores. There are conflicting results regarding the relationship between PD-L1 expression and TMB [34,35,52,53,54]. This discrepancy may be due to the anti-PD-L1 IHC antibodies used in the assessment. A previous study used the PD-L1 antibody clone SP142, which showed lower levels of PD-L1 expression compared with the PD-L1 22C3 assay used in our study [55,56]. In addition, there is a possibility for the existence of different PD-L1 expression among the tumor types; further analysis of this aspect is required to validate the relationship between PD-L1 expression and TMB in different cohorts of various tumors.

MSI-H carcinomas are associated with a high-TMB [57]. In our study, the MSI score correlated with the TMB values and PD-L1 expression. In addition to the MSI status as high and low, the MSI score may also provide helpful information and warrants further study.

## 5. Conclusions

PD-L1 expression is significantly associated with TMB and MSI score and this correlation depends on the location of the primary tumor. Further studies will be needed to better clarify their potential roles in ICI therapy in various cancers.

## Figures and Tables

**Figure 1 cancers-13-04659-f001:**
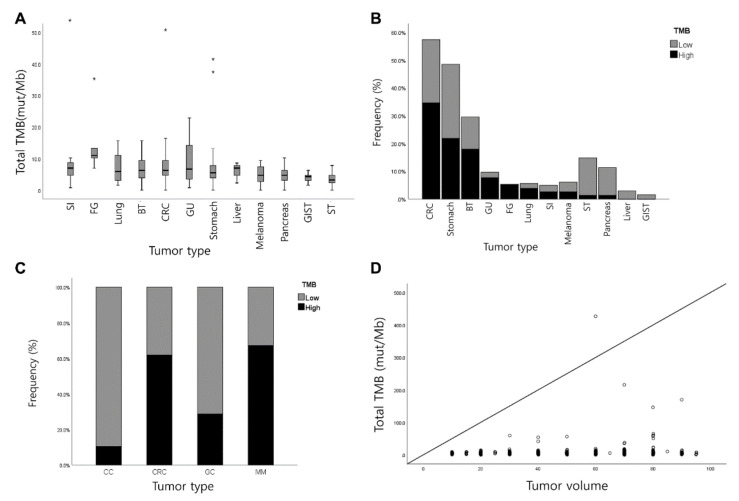
Tumor mutation burden (TMB) results in 588 cases. (**A**) TMB values across various tumor types. The highest average TMB was observed in the small intestine, followed by that in the female genital tract, lung cancer, biliary tract cancer, CRC, genitourinary tract cancer, GC, liver cancer, MM, GIST, and mesenchymal tumor. (**B**) The proportion of high-TMB (> 10 mut/MB) among tumor types in a total of 588 cases and (**C**) previously reported TCGA data. (**D**) Correlation between tumor volume and TMB. (*, outliers, Abbreviations: TMB, tumor mutation burden; SI, small intestine; FG, female genital tract, CRC, colorectal cancer; BT, biliary tract cancer; GC, gastric cancer; CC, cholangiocarcinoma; MM, melanoma; GIST, gastrointestinal stromal tumor; ST, soft tissue tumor).

**Figure 2 cancers-13-04659-f002:**
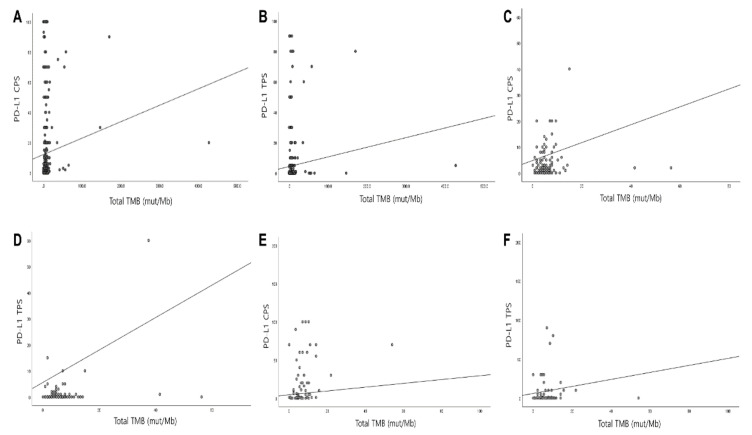
Correlation of the PD-L1 score with the tumor mutation burden (TMB). (**A**) Correlation of TMB and PD-L1 CPS score and in (**B**) TPS scores all samples. Significant correlation of (**C**) TMB and PD-L1 CPS score and (**D**) TMB and PD-L1 TPS score in gastric carcinoma. Significant correlation of (**E**) TMB and PD-L1 CPS score and (**F**) TMB and PD-L1 TPS score in biliary tract tumor. (Abbreviations: TMB, tumor mutation burden; CPS, combined positive score; TPS, tumor proportion score).

**Figure 3 cancers-13-04659-f003:**
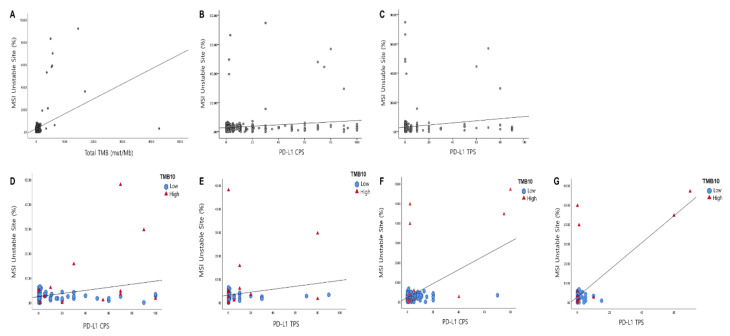
Correlation of microsatellite instability (MSI) site percentage with tumor mutation burden (TMB) and PD-L1 expression. MSI site percentage showed a significant correlation with (**A**) TMB, (**B**) PD-L1 CPS score, and (**C**) PD-L1 TPS score in 588 cases. The MSI site percentage was significantly correlated with the PD-L1 CPS (**D**) (*p* = 0.014, R^2^ = 0.081), but not with PD-L1 TPS (**E**) in biliary tract carcinoma (*p* = 0.118). In gastric carcinoma, the MSI site percentage was significantly correlated with PD-L1 CPS score (**F**) and PD-L1 TPS score (**G**) (*p* < 0.001 and R^2^ = 0.291; *p* < 0.001 and R^2^ = 0.506, respectively).

**Table 1 cancers-13-04659-t001:** Clinicopathologic features of 588 samples.

Category	Variables	Number of Cases (*n* = 588, %)
Age (years)	Mean ± SD (Range, median)	58.26 ± 11.84 (18–87, 59)
Sex	Female	230 (39.1)
	Male	358 (60.9)
Source of tissue	Our institute	534 (90.8)
	Other institutes	54 (9.2)
Paraffin block age (days)	Mean ± SD (Range, median)	250.72 ± 439.11 (2–3536, 53)
Pathologic diagnosis	Gastric cancer	153 (26.0)
	Tubular adenocarcinoma	48
	Poorly cohesive carcinoma	97
	Papillary adenocarcinoma	2
	Mucinous adenocarcinoma	2
	Medullary carcinoma with lymphoid stroma	1
	Squamous cell carcinoma	1
	Neuroendocrine carcinoma	1
	Mixed neuroendocrine-nonneuroendocrine neoplasm	1
	Colorectal cancer	143 (24.3)
	Adenocarcinoma	140
	Mucinous adenocarcinoma	1
	Undifferentiated carcinoma	1
	Neuroendocrine carcinoma	1
	*Malignant melanoma*	20 (3.4)
	Cutaneous	3
	Acral and subungal	8
	Mucosal	8
	Unknown	1
	GIST *	8 (1.4)
	Small intestine	5
	Stomach	1
	Extragastrointestinal	2
	Hepatobiliary carcinoma	88 (15.0)
	Cholangiocarcinoma	47
	Hepatocellular carcinoma	11
	Combined hepatocellular carcinoma and cholangiocarcinoma	2
	Adenocarcinoma, NOS	19
	Adenosquamous carcinoma	2
	Intracystic papillary neoplasm with associated invasive carcinoma	1
	Carcinoma, undifferentiated, NOS	4
	Neuroendocrine carcinoma, NOS	1
	Epithelioid hemangioendothelioma	1
	Pancreas carcinoma	52 (8.8)
	Ductal adenocarcinoma	47
	Undifferentiated carcinoma with osteoclast-like giant cells	2
	Acinar cell carcinoma	1
	Neuroendocrine tumor, NOS	2
	Small intestine carcinoma	14 (2.4)
	Adenocarcinoma, NOS	14
	Kidney and Genitourinary tract cancer	16 (2.7)
	Infiltrating urothelial cell carcinoma	15
	Renal cell carcinoma	1
	Mesenchymal tumor	69 (11.7)
	Alveolar soft part sarcoma	2
	Angiosarcoma	5
	Chondrosarcoma	1
	Clear cell sarcoma	1
	Dedifferentiated liposarcoma	14
	Epithelioid sarcoma	2
	Ewing sarcoma	5
	Extraskeletal myxoid chondrosarcoma	2
	Intimal sarcoma	1
	Leiomyosarcoma	12
	Malignant peripheral nerve sheath tumor	1
	Malignant perivascular epithelioid cell neoplasm	1
	Mesenchymal chondrosarcoma	1
	Myxofibrosarcoma	1
	Myxoid liposarcoma	3
	Osteosarcoma	1
	Primary intimal sarcoma	1
	Rhabdomyosarcoma	1
	Solitary fibrous tumor	4
	Synovial sarcoma	1
	Undifferentiated pleomoprhic sarcoma	2
	Undifferentiated pleomorphic sarcoma	4
	Undifferentiated spindle cell sarcoma	2
	Well differentiated liposarcoma	1
	Female genital tract cancer	6 (1.0)
	Ovarian serous carcinoma	2
	Squamous cell carcinoma, uterine cervix	1
	Endometrioid adenocarcinoma, uterine corpus	1
	Endometrial stromal sarcoma	1
	Adenosarcoma, uterine corpus	1
	Lung cancer	12 (2.0)
	Adenocarcinoma	6
	Mucinous adenocarcinoma	2
	Squamous cell carcinoma	2
	Combined large cell neuroendocrine carcinoma	1
	Small cell neuroendocrine carcinoma	1
	Other carcinoma	7 (1.2)
	Adrenocortical carcinoma	2
	Extramammary Paget disease	1
	Appendiceal goblet cell adenocarcinoma	1
	Appendiceal signet ring cell adenocarcinoma	1
	Mucinous adenocarcinoma of retroperitoneum	1
	Unknown primary	1
Primary vs. metastasis	Primary	478 (81.3)
	Metastasis	110 (18.7)
Specimen type	Biopsy	289 (49.1)
	Resection	299 (50.9)

Abbreviations: SD, standard deviation; GIST, gastrointestinal stromal tumor; * Negative.

**Table 2 cancers-13-04659-t002:** Correlation of PD-L1 combined positive score (CPS) with tumor mutation burden (TMB) according to primary tumor.

**PD-L1 CPS ***	**Total (*n* = 413)**				**Gastric Cancer (*n* = 126)**				**Biliary Tract Cancer (*n* = 73)**			
	Negative (*n* = 105)	Low (*n* = 265)	High (*n* = 43)	*p*-value	Negative (*n* = 41)	Low (*n* = 82)	High (*n*=2)	*p*-value	Negative (*n* = 15)	Low (*n* = 44)	High (*n* = 14)	*p*-value
TMB < 10	100 (95.2%)	230 (86.8%)	33 (76.6%)	0.005	40 (97.6%)	73 (89.0%)	1 (33.3%)	0.001	14 (93.3%)	37 (84.1%)	8 (57.1%)	0.032
TMB ≥ 10	5 (4.8%)	35 (13.2%)	10 (23.3%)		1 (2.4%)	9 (11.0%)	2 (66.7%)		1 (6.7%)	7 (15.9%)	6 (42.9%)	
**PD-L1 CPS ****	**Total (*n* = 413)**				**Gastric Cancer (*n* = 126)**				**Biliary Tract Cancer (*n* = 73)**			
	Negative (*n* = 107)	Positive (*n* = 306)		*p*-value	Negative (*n* = 41)	Positive (*n* = 85)		*p*-value	Negative (*n* = 16)	Positive (*n* = 57)		*p*-value
TMB < 10	101 (94.4%)	262 (85.6%)		0.017	40 (97.6%)	74 (87.1%)		0.060	15 (93.8%)	44 (77.2%)		0.137
TMB ≥ 10	6 (5.6%)	44 (14.4%)			1 (2.4%)	11 (12.9%)			1 (6.2%)	13 (22.8%)		

* Negative: PD-L1 expression < 1, Low: 1 ≤ PD-L1 expression < 50, High: 50 ≤ PD-L1 expression ** Negative: PD-L1 expression < 1, Positive: 1 ≤ PD-L1 expression.

**Table 3 cancers-13-04659-t003:** Correlation of PD-L1 tumor proportion score (TPS) with tumor mutation burden (TMB) according to primary tumor.

PD-L1 TPS *	Total (*n* = 413)				Gastric Cancer (*n* = 126)				Biliary Tract Cancer (*n* = 73)			
	Negative (*n* = 289)	Low (*n* = 106)	High (*n* = 18)	*p*-value	Negative (*n* = 98)	Low (*n* = 26)	High (*n* = 2)	*p*-value	Negative (*n* = 46)	Low (*n* = 23)	High (*n* = 4)	*p*-value
TMB < 10	257 (88.9%)	92 (86.8%)	14 (77.8%)	0.343	91 (92.9%)	23 (88.5%)	0 (0.0%)	<0.001	38 (82.6%)	19 (82.6%)	2 (50.0%)	0.273
TMB ≥ 10	32 (11.1%)	14 (13.2%)	4 (22.2%)		7 (7.1%)	3 (11.5%)	2 (100.0%)		8 (17.4%)	4 (17.4%)	2 (50.0%)	

* Negative: PD-L1 expression < 1, Low: 1 ≤ PD-L1 expression < 50, High: 50 ≤ PD-L1 expression.

**Table 4 cancers-13-04659-t004:** Correlation of PD-L1 CPS, PD-L1 TPS, and TMB with MSI status in entire cases.

	CPS *			TPS *				TMB		
	Negative (<1) (*n* = 107)	Positive (≥1) (*n* = 306)	*p*-value	Negative (<1) (*n* = 290)	Low (1–49) (*n* = 105)	High (≥50) (*n* = 18)	*p*-value	TMB-L (<10) (*n* = 363)	TMB-H (≥10) (*n* = 51)	*p*-value
MSI-L	107 (100%)	298 (97.4%)	0.091	286 (98.6%)	104 (99.0%)	15 (83.3%)	<0.001	363 (100.0%)	43 (84.3%)	<0.001
MSI-H	0 (0.0%)	8 (2.6%)		4 (1.4%)	1 (1.0%)	3 (16.7%)		0 (0.0%)	8 (15.7%)	

Abbreviation: CPS, combined positive score; TPS, tumor proportion score; TMB, tumor mutation burden; MSI, microsatellite instability; L, low; H, high * All 414 case were available for PD-L1 study, however one case was not unable to evaluate due to limited number of tumor cells.

## Data Availability

All data and materials are available from the authors upon reasonable request by contacting K.-M.K. (kkmkys@skku.edu).

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
