# Peer review of "PD-L1 Expression Is Significantly Associated with Tumor Mutation Burden and Microsatellite Instability Score"

_cancers, 2021, doi:10.3390/cancers13184659_

Round 1

Reviewer 1 Report

This manuscript addressed the association of several factors in predicting the response to immunotherapy. Samples from 588 cancer patients were included in this study for IHC or sequencing and 7 for whole-exome sequencing (WES) to validate the results.

Here are a few comments:

  1. The definition of biomarker, TMB and MSI? Since many types of genomic alterations were mentioned later in this manuscript - such as CNVs, SNVs, indels, fusions - I think it will be good to clearly define the terms at the beginning.
  2. The significance and innovation of the study could be explained a bit better in the introduction. Now the logic flow is like: 1) How to predict immunotherapeutic response is missing. 2) PD-L1 expression is a therapeutic biomarker of response to immune checkpoint blockade (ICB). 3) Same for TMB and MSI. 4) Measuring TMB could be superior to neoantigen detection. 5) MSI can be detected by PCR or NGS. 6) Here PD-L1 expression is compared with TMB and MSI in cancer patients using a commercial panel.

What goal the authors wanted to achieve here? I think some background/connection is missing to make it clear. For instance, the 588 patients recruited here were under chemotherapy treatment, not immunotherapy. (Actually, no immunotherapy was mentioned as all except for the introduction.) Also having a brief conclusion/rationale would be helpful in the last paragraph.

  1. In the introduction the authors advocate for TMB assessment over neoantigens detection. According to result section 3.3, while the sequencing size is 1.28 Mbps in TSO 500 and 35.9 Mbps in WES, TMB values measured by the TSO 500 and WES methods are comparable. Presumably this is the point the authors wanted to address. If so, the neoantigen part can be moved to discussion or else it sounds confusing.
  2. In result section 3.1, the researchers compared 80ng with 120ng DNA input in the sequencing overage. However, I noticed in the method section 2.2 that 40ng of DNA was used as input for the library preparation. So how is the sequencing coverage of that?
  3. The mutation rate varies substantially among different types of tumors, as shown in the manuscript (TMB ranging from 0-426.8). Even within the same type of cancer, the variance is large ( Small intestine TMB, 36.31 ± 112.42 mut/Mb). So, use the median value may be not helpful in setting up any correlation. Why not to analyze the high-TMB and low-TMB groups separately?
  4. It seems that the age of paraffin blocks (>7 years) has a significant effect on IHC detection of ER and PR in breast cancer (H. Chen, Q. Fang, B. Wang. Oncology Letters. 2020). Is there a similar concern for PD-L1 IHC detection? I saw the paraffin age ranges from 2 to 3536 in the method.
  5. In Fig 2, most samples have TMB close to 0 or in the lower range and the Spearman’s correlation coefficient seems quite low. Again, high-TMB and low-TMB groups could be examined separately here as well. This might explain why TMB failed to show significance in many organs here. Additional curiosity is, why to use Spearman’s correlation instead of Pearson's correlation here?
  6. In Fig 3E, the most prominent example of biliary tract carcinoma showed no correlation of MSI scores and PD-L1 expression. Isn’t this weakening the conclusion of this manuscript?
  7. In result section 3.5, it’s good to identify these novel fusions from patients. But what are the correlation with PD-L1? Is it a bit off topic here?
  8. The same thing for Fig 5, c-MET was mentioned out of nowhere. How this links to PD-L1? Having a smooth transition here will be better.
  9. Texts/axis in the figures are very tiny and blurry.

Reference:

Chen, H., Fang, Q. Q., & Wang, B. (2020). The age of paraffin block influences biomarker levels in archival breast cancer samples. Oncology letters20(1), 525–532. https://doi.org/10.3892/ol.2020.11586

Author Response

Reviewer 1

 Comments and Suggestions for Authors

This manuscript addressed the association of several factors in predicting the response to immunotherapy. Samples from 588 cancer patients were included in this study for IHC or sequencing and 7 for whole-exome sequencing (WES) to validate the results.

Here are a few comments:

The definition of biomarker, TMB and MSI? Since many types of genomic alterations were mentioned later in this manuscript - such as CNVs, SNVs, indels, fusions - I think it will be good to clearly define the terms at the beginning.

Response: In accordance with the reviewer’s suggestion, we added the definition of TMB and MSI in the Introduction and definitions of CNVs, SNVs, indels, and fusions in Methods 2.3 and 2.5 (pages 2, 3, and 4, highlighted).

The significance and innovation of the study could be explained a bit better in the introduction. Now the logic flow is like: 1) How to predict immunotherapeutic response is missing. 2) PD-L1 expression is a therapeutic biomarker of response to immune checkpoint blockade (ICB). 3) Same for TMB and MSI. 4) Measuring TMB could be superior to neoantigen detection. 5) MSI can be detected by PCR or NGS. 6) Here PD-L1 expression is compared with TMB and MSI in cancer patients using a commercial panel.

What goal the authors wanted to achieve here? I think some background/connection is missing to make it clear. For instance, the 588 patients recruited here were under chemotherapy treatment, not immunotherapy. (Actually, no immunotherapy was mentioned as all except for the introduction.) Also having a brief conclusion/rationale would be helpful in the last paragraph.

Response: To determine the relationship between biomarkers for patient stratification in the immunotherapy of patients with advanced solid tumors, we compared PD-L1 IHC results with TMB values and the percentage of unstable microsatellite loci using a commercially available comprehensive cancer panel assay (CCPA) with >500 genes. We added such aims more clearly in the last paragraph of the revised Introduction as suggested (page 2, fourth paragraph, highlighted).

In the introduction the authors advocate for TMB assessment over neoantigens detection. According to result section 3.3, while the sequencing size is 1.28 Mbps in TSO 500 and 35.9 Mbps in WES, TMB values measured by the TSO 500 and WES methods are comparable. Presumably this is the point the authors wanted to address. If so, the neoantigen part can be moved to discussion or else it sounds confusing.

Response: In accordance with the reviewer’s suggestion, we moved the neoantigen section from Introduction to Discussion (page 9, second paragraph, highlighted).

In result section 3.1, the researchers compared 80ng with 120ng DNA input in the sequencing overage. However, I noticed in the method section 2.2 that 40ng of DNA was used as input for the library preparation. So how is the sequencing coverage of that?

Response: The minimum amount of DNA input recommended by the manufacturer was 40 ng. The average depth of sequencing coverage was ×150-300 with a 40 ng input amount of DNA. However, to ensure sufficient and stable coverage, we altered the DNA input amount, so 120 ng of DNA was used for TSO 500. The corrected methods are described in section 2.2 (page 3, highlighted).

It seems that the age of paraffin blocks (>7 years) has a significant effect on IHC detection of ER and PR in breast cancer (H. Chen, Q. Fang, B. Wang. Oncology Letters. 2020). Is there a similar concern for PD-L1 IHC detection? I saw the paraffin age ranges from 2 to 3536 in the method.

Response: PD-L1 staining might decrease, especially with a sample old more than 3 years [1,2]. In our cohort, block ages in most cases were less than 3 years, and there were 15 cases with a block age of > 3 years, but less than 5 years, and there was no significant difference in PD-L1 positivity between these blocks. Therefore, we assumed that block age did not affect PD-L1 expression positivity.

The mutation rate varies substantially among different types of tumors, as shown in the manuscript (TMB ranging from 0-426.8). Even within the same type of cancer, the variance is large (Small intestine TMB, 36.31 ± 112.42 mut/Mb). So, use the median value may be not helpful in setting up any correlation. Why not to analyze the high-TMB and low-TMB groups separately?

In Fig 2, most samples have TMB close to 0 or in the lower range and the Spearman’s correlation coefficient seems quite low. Again, high-TMB and low-TMB groups could be examined separately here as well. This might explain why TMB failed to show significance in many organs here. Additional curiosity is, why to use Spearman’s correlation instead of Pearson's correlation here?

Response: In accordance with the reviewer’s suggestion, we categorized TMB with a cut-off value of 10 mut/Mb and performed a chi-squared test and Fisher’s exact test for PD-L1 IHC expression. In addition, statistical analyses were performed with Pearson’s correlation, and new results have been added in Results 3.4 (page 6) and a new Table 2 and Table 3.

In Fig 3E, the most prominent example of biliary tract carcinoma showed no correlation of MSI scores and PD-L1 expression. Isn’t this weakening the conclusion of this manuscript?

Response: Of 73 biliary tract carcinomas, only two cases showed high MSI. The lack of correlation between MSI and PD-L1 expression might be caused by the small number of MSI-high cases.

In result section 3.5, it’s good to identify these novel fusions from patients. But what are the correlation with PD-L1? Is it a bit off topic here?

The same thing for Fig 5, c-MET was mentioned out of nowhere. How this links to PD-L1? Having a smooth transition here will be better.

Response: As Reviewer #2 also suggested taking out these descriptions because they are not related to this manuscript’s topic, we decided to remove this part to make this paper more fluid for the readers.

Texts/axis in the figures are very tiny and blurry.

Response: The figures have been modified. Thank you for your comments.

Reviewer 2 Report

This is a study of 588 cases of select cancer types where the authors examined the PD-L1 expression and its correlation with tumor mutational burden and microsatellite instability.  In addition, the authors present data on novel fusions as well as MET splicing variant confirmed by c-MET overexpression. 

Major Comments:

  1. The main concern for this manuscript is the methodology used to compare PD-L1, TMB, and MSI.  Specifically, PD-L1 IHC should be compared as a categorical variable instead of a continuous variable because pathologist score PD-L1 IHC based on cut-offs.  For example, for DAKO 22C3, the current companion diagnostics cutoff is a TPS of 1 with a previous cut-off of TPS 50.  In some tumor types such as cervical cancer a CPS 1 cutoff is used, and in others such as ESCC a CPS 10 is used.  In sum, due to the way pathologist score these PD-L1 cases, treating PD-L1 IHC as a continuous variable is not appropriate.  
  1. After using a categorical variable as a cut-off, should compare this data set to PMID 32884129. In this paper, the authors examined 48k cases of advanced cancer patient’s teste with PD-L1 IHC and comprehensive genomic profiling and discovered that from a clinically relevant perspective (based on CDx cutoffs for PD-L1, TMB, and MSI) PD-L1 IHC was highly correlated with CD274 amplification but not quite so with TMB and MSI.  However, similar to the authors of this current paper under review, TMB and MSI was highly correlated with each other. 

Minor Comments:

  1. The novel fusion section and MET splicing variant sections could potentially be taken out as it does not relate to the main point of the paper which is correlation of PD-L1 IHC with TMB and MSI. 

Author Response

Reviewer 2

Comments and Suggestions for Authors

This is a study of 588 cases of select cancer types where the authors examined the PD-L1 expression and its correlation with tumor mutational burden and microsatellite instability.  In addition, the authors present data on novel fusions as well as MET splicing variant confirmed by c-MET overexpression.

Major Comments:

The main concern for this manuscript is the methodology used to compare PD-L1, TMB, and MSI.  Specifically, PD-L1 IHC should be compared as a categorical variable instead of a continuous variable because pathologist score PD-L1 IHC based on cut-offs.  For example, for DAKO 22C3, the current companion diagnostics cutoff is a TPS of 1 with a previous cut-off of TPS 50.  In some tumor types such as cervical cancer a CPS 1 cutoff is used, and in others such as ESCC a CPS 10 is used.  In sum, due to the way pathologist score these PD-L1 cases, treating PD-L1 IHC as a continuous variable is not appropriate. 

Response: In accordance with the reviewer’s critical comments, PD-L1 expression was categorized into two- or three-tiered systems according to previous reports [3,4]. We performed the chi-square test and Fisher’s exact test to evaluate the correlation between PD-L1 IHC and TMB, and all the new results have been added in Results 3.4 (page 6) and a new Table 2 and Table 3.

After using a categorical variable as a cut-off, should compare this data set to PMID 32884129. In this paper, the authors examined 48k cases of advanced cancer patient’s teste with PD-L1 IHC and comprehensive genomic profiling and discovered that from a clinically relevant perspective (based on CDx cutoffs for PD-L1, TMB, and MSI) PD-L1 IHC was highly correlated with CD274 amplification but not quite so with TMB and MSI.  However, similar to the authors of this current paper under review, TMB and MSI was highly correlated with each other.

Response: We have read Huang et al.’s paper with great interest. They reported a high correlation between PD-L1 expression and CD274 gene amplification (p < 0.0001), but they also found significant correlations between PD-L1 expression and MSI and TMB (p < 0.0001). With different cut-off values of PD-L1, we re-evaluated the association of PD-L1 IHC with MSI and TMB; the new results have been added in Results section 3.4 and a new Table 4, and Huang et al.’s paper [5] has been added as a reference in the Discussion section.

Minor Comments:

The novel fusion section and MET splicing variant sections could potentially be taken out as it does not relate to the main point of the paper which is correlation of PD-L1 IHC with TMB and MSI.

Response: In accordance with the reviewer’s suggestion, we have removed these findings from the revised manuscript.

Round 2

Reviewer 2 Report

Manuscript has responded adequately to previous Review.